# CamPilot: Improving Camera Control in Video Diffusion Model with Efficient Camera Reward Feedback

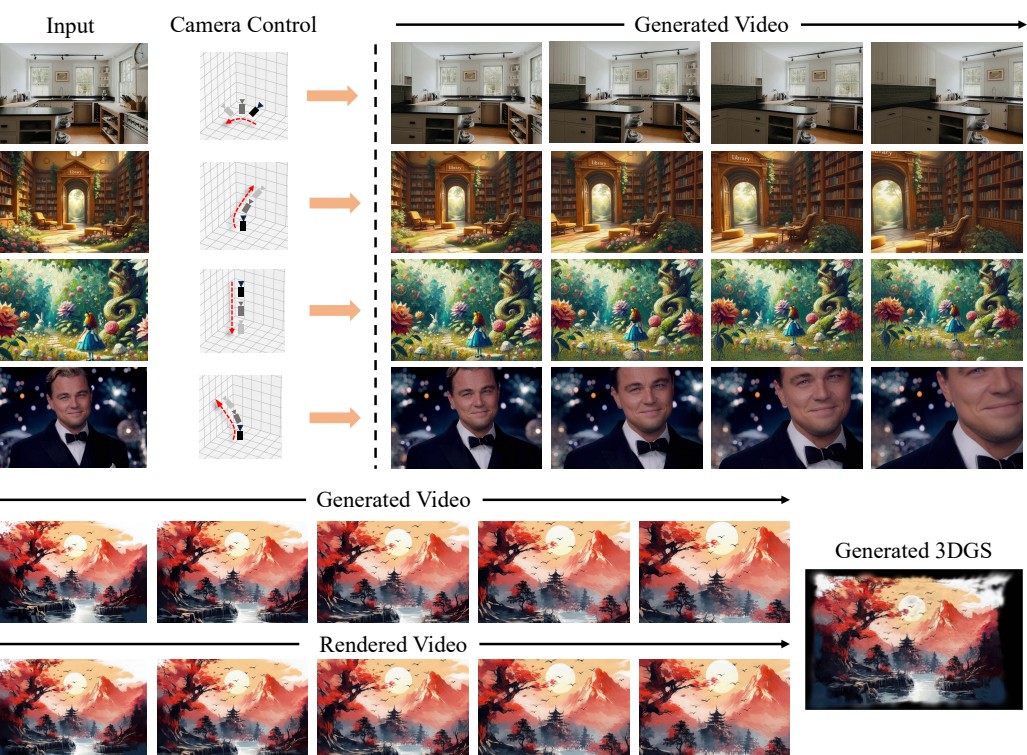

Figure 1: Our model functions as a comprehensive framework for world-consistent video generation and scene reconstruction. In the upper section, it excels at generating 3D-consistent scene videos for world exploration by following custom camera trajectories. In the lower section, it efficiently reconstructs high-quality 3D scenes in a feed-forward manner with generated video frames.

## ABSTRACT

Recent advancements in camera-controlled video diffusion models have significantly improved video-camera alignment and enabled more accurate 3D scene generation, driven by potential downstream applications such as virtual reality. However, we reveal that existing approaches often struggle to precisely adhere to the given camera conditions, leading to inconsistencies in the 3D geometry. Inspired by Reward Feedback Learning in diffusion models, which has demonstrated strong potential in aligning model outputs with task-specific objectives, we build upon this paradigm and aim to further improve camera controllability. Directly borrowing existing ReFL approaches faces several challenges. First, current reward models lack the capacity to assess video-camera alignment. Second, decoding latent into RGB videos for reward computation introduces substantial computational overhead. Third, 3D geometric information is typically neglected during video decoding. To address these limitations, we introduce a camera-aware 3D decoder that efficiently decodes video latent into 3D representations for reward computation. Specifically, we project the video latent and camera pose into 3D Gaussians, which supports efficient rendering from arbitrary views. In this pro-

cess, the camera pose not only acts as an input variable but also serves as a projection parameter for determining the mean of each 3D Gaussian. If the generated video does not match the camera conditions, the 3D structure becomes geometrically inconsistent, leading to blurry rendered images. Based on this property, we explicitly optimizing pixel-level consistency between rendered novel views and ground-truth ones as reward feedback. To accommodate the stochastic nature, we further introduce a visibility term that selectively supervises only deterministic regions derived via geometric warping. Extensive experiments conducted on the RealEstate10K and WorldScore benchmarks demonstrate the effectiveness of our proposed method in enhancing both camera controllability and generation quality.

# 1 INTRODUCTION

Video diffusion models have recently achieved impressive progress (Blattmann et al., 2023; Yang et al., 2024c; Liu et al., 2024), enabling the generation of high-quality and temporally coherent videos conditioned on inputs such as text prompts or a single image. Despite these advances, real-world applications often demand a higher degree of controllability. A key factor is camera controllability. Users not only expect visually realistic content but also require explicit control over camera trajectories to support user-friendly and customizable content creation.

To address the need for camera-controlled video generation, several recent works (Yu et al., 2024; Ren et al., 2025; Gao et al., 2024; Sun et al., 2024; Voleti et al., 2024; Chan et al., 2023; Sargent et al., 2024; Bahmani et al., 2024a; He et al., 2024) have explored this task by fine-tuning pretrained video models with paired camera conditioning. Recognizing that many downstream applications such as virtual reality (Schuemie et al., 2001), robotics (Mateo et al., 2016), and game development (Gregory, 2018) require not only high-quality visuals but also consistent 3D representations, these methods have begun to bridge the gap between 2D generation and 3D reconstruction. A common strategy is to reconstruct 3D by optimizing over generated novel views. Despite these advancements, precise camera control is still difficult to achieve in practice, often resulting in inconsistent and suboptimal convergence during 3D reconstruction. In fact, improving the alignment between generated content and given conditions is a long-standing problem in generative models.

Recent works (Prabhudesai et al., 2024; Li et al., 2024b; Liu et al., 2025; Zhang et al., 2024a; Xu et al., 2023; Prabhudesai et al., 2023) have introduced Reward Feedback Learning (ReFL) for diffusion models to further refine the model according to human preferences or task-specific objectives, drawing inspiration from the Reinforcement Learning from Human Feedback (RLHF) (Grattafiori et al., 2024; Yang et al., 2024a; Lee et al., 2023) of large language models (LLMs). For instance, VADER (Prabhudesai et al., 2024) explores a range of reward functions—such as perceptual quality, text-video semantic alignment, and aesthetic appeal—to enhance visual fidelity and semantic consistency. Controlnet++ (Li et al., 2024b) leverages pixel-level cycle consistency as a reward to improve image-based controllability. However, none of these approaches considers camera controllability.

In this work, we aim to enhance the adherence to camera conditioning through ReFL, a topic that remains under-explored in the context of video diffusion. However, there are three main challenges in adopting this strategy for camera-controlled video diffusion. First, current models struggle to assess the alignment of camera conditions in video generation. Second, reward computation necessitates decoding the generated latent into video, leading to VRAM inefficiency due to the resource-intensive nature of video decoders. Lastly, these methods often overlook the underlying 3D geometric structure during video decoding, which restricts their effectiveness in the 3D-like task. A naive approach would be to use COLMAP (Schönberger & Frahm, 2016) for camera pose estimation. However, the heavy computational cost and scale-invariant pose estimation make it infeasible for efficient training and precise pose supervision. Considering the three challenges, we introduce a camera-aware 3D decoder that enables computationally efficient evaluation of video-camera consistency without requiring heavy computation. Specifically, we project the video latent—obtained by encoding a raw video using the video VAE—along with the corresponding ground-truth camera poses into a 3D representation, namely 3D Gaussians (3DGS) (Kerbl et al., 2023). This representation supports efficient novel view rendering from arbitrary viewpoints and utilizes photometric loss for supervision. In this projection process, camera poses play a crucial role. On the one hand, they are transformed into Plücker embeddings (He et al., 2024) as part of the network input. On the other hand, the mean

of each 3D Gaussian is computed by projecting the camera pose along with the predicted depth. These two mechanisms ensure that when the generated video latent is misaligned with the input camera poses, the resulting 3DGS becomes geometrically inconsistent, leading to degraded renderings. Based on this property, we regard minimizing the pixel-level difference between the rendered videos and ground-truth sequences as a camera-aware reward. This design is consistent with the nature of the proposed camera-aware 3D decoder, which emphasizes low-level visual cues.

However, computing pixel-level rewards presents unique challenges. High-level semantic rewards can be meaningfully applied across multiple diverse diffusion samples, while low-level pixel alignment rewards are sensitive to diverse generation results. Camera-controlled video generation often involves hallucinated content, making it difficult to enforce strict pixel-level consistency across all pixels without suppressing generative diversity. To address this, our reward formulation is carefully designed to focus only on deterministic regions that are visible in the conditioning image, while ignoring unconstrained areas that permit creative generation. To this end, we design a visibility-aware reward objective that restricts reward computation to deterministic regions while avoiding penalization in hallucinated or occluded areas. Since our camera-aware 3D decoder is inherently 3D-aware, we can render depth maps from the 3DGS. By combining the rendered depth with camera poses, we can determine the visibility of each pixel across all frames through geometric warping.

To summarize, our contributions are listed as follows.

- We propose a camera-aware 3D decoder that lifts the video latent along with the camera pose into 3DGS, which supports efficient rendering from arbitrary viewpoints and enables the evaluation of the alignment between camera conditions and the generated video.
- We employ reward-based feedback learning to further improve the alignment between the video and the camera by regarding the minimization of the deterministic pixel-level difference between the rendered videos and ground-truth videos as a reward.
- Extensive experiments demonstrate the effectiveness of the proposed framework, significantly improving camera controllability and visual quality.

## 2 RELATED WORK

### 2.1 CAMERA CONTROLLED VIDEO DIFFUSION MODELS

With the rapid advancements in video diffusion models, camera-controlled video generation (He et al., 2024; Bahmani et al., 2024b;a; Yu et al., 2024; Ren et al., 2025; Gao et al., 2024; Wang et al., 2024b; Hu et al., 2025) has garnered significant attention in the research community. Recent works such as MotionCtrl (Wang et al., 2024b), CameraCtrl (He et al., 2024), and ViewCrafter (Yu et al., 2024) inject various forms of camera conditioning—ranging from extrinsics and Plücker embeddings (Sitzmann et al., 2021) to point cloud renders—into pretrained video generation models. More recently, AC3D (Bahmani et al., 2024a) has carefully explored the spatial and temporal points at which camera representations should be injected. CameraCtrl2 (He et al., 2025) investigates this task from a dataset curation perspective to enable dynamic scene generation with controllable cameras. CamCo (Xu et al., 2024a) introduces epipolar constraints into attention layers, while Gen3C (Ren et al., 2025) and FlexWorld (Chen et al., 2025) maintain a spatiotemporal 3D cache to enhance robustness in camera control. Despite these advances, existing approaches still face challenges in achieving precise control and remain largely constrained to 2D video generation. In this work, we enhance camera controllability through reward feedback learning and, importantly, enable the simultaneous generation of corresponding 3D counterparts in an efficient feed-forward manner. Our framework adopts Plücker embeddings as the camera condition. However, the proposed preference fine-tuning is a general method and can be applied to any form of camera condition representation.

### 2.2 3D GENERATIVE MODELS

Object-level 3D generative models (Hong et al., 2023; Zhang et al., 2024b; Ge et al., 2024; 2023; Zhang et al., 2024c; Jiang et al., 2025; Xu et al., 2024b) have made remarkable progress in recent years, largely driven by the availability of large-scale 3D object datasets. However, 3D scene generation remains relatively under-explored. Most video diffusion based approaches (Yu et al., 2024; Ren et al., 2025; Gao et al., 2024; Sun et al., 2024; Voleti et al., 2024; Chan et al., 2023; Sargent

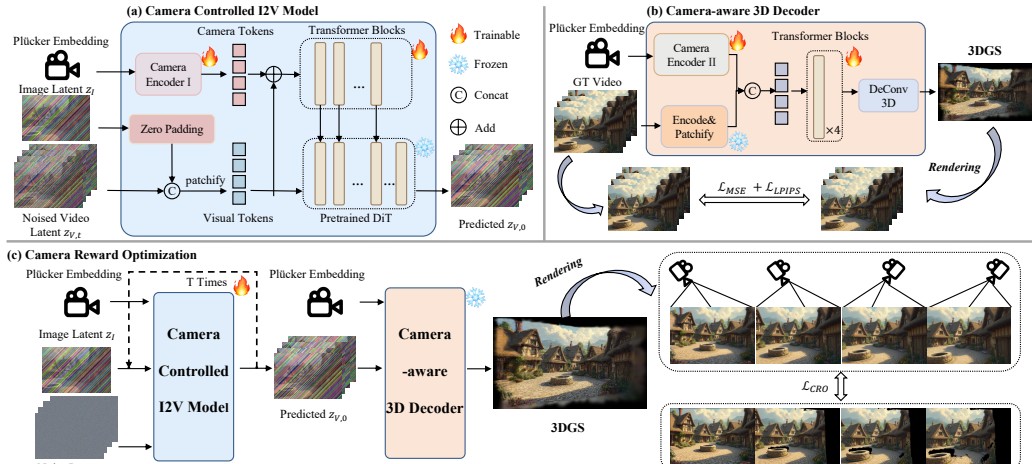

Figure 2: Overall of our framework. It consists of (a): a camera-controlled I2V model, where we inject Plücker Embedding as camera condition using ControlNet. (b) A camera-aware 3D decoder that decodes latent to 3DGS, supporting rendering for reward computation. (c) Camera reward optimization that minimizes mask-aware difference between rendered videos and ground-truth ones.

et al., 2024) typically adopt a two-stage pipeline. In the first stage, diffusion models are employed to generate novel views given sparse or single-view observations and target poses. In the second stage, per-scene optimization is conducted using the generated novel views and corresponding target poses. Object-level 3D generative models (Hong et al., 2023; Zhang et al., 2024b; Ge et al., 2024; Li et al., 2025; Zhang et al., 2024c; Jiang et al., 2025) have made remarkable progress in recent years, largely driven by the availability of large-scale 3D object datasets. However, 3D scene generation remains relatively under-explored. Most video diffusion based approaches (Yu et al., 2024; Ren et al., 2025; Gao et al., 2024; Sun et al., 2024; Voleti et al., 2024; Chan et al., 2023; Sargent et al., 2024) typically adopt a two-stage pipeline. In the first stage, diffusion models are employed to generate novel views given sparse or single-view observations and target poses. In the second stage, per-scene optimization is conducted using the generated novel views and corresponding target poses. Despite their effectiveness, such two-stage approaches suffer from two main limitations. First, the per-scene optimization process is time-consuming, making it difficult to scale to large numbers of scenes. Second, the quality of scene reconstruction is highly sensitive to the consistency between the generated novel views and the target camera poses. misalignment between them can lead to suboptimal convergence. In contrast, we propose a camera-aware 3D decoder that not only enables efficient 3D scene reconstruction in a feed-forward manner, but also serves as a reward function to minimize the misalignment between generated novel views and their corresponding target poses.

## 2.3 ALIGNING DIFFUSION MODELS WITH PREFERENCE

Drawing inspiration from Reinforcement Learning from Human Feedback (RLHF) in the field of large language models (LLMs), recent works have begun to incorporate similar paradigms into diffusion models to better align generation quality with human preferences (Yang et al., 2024b; Prabhudesai et al., 2024; Yuan et al., 2024; Liu et al., 2025; Li et al., 2024a;b; Xu et al., 2023; Zhang et al., 2024a). For instance, ControlNet++ (Li et al., 2024b) explicitly optimizes pixel-level cycle consistency between generated images and conditional controls for improving controllable generation. UniFL Zhang et al. (2024a) proposes a unified framework that leverages feedback learning to enhance diffusion models comprehensively. VADER (Prabhudesai et al., 2024) explores a variety of reward models to fine-tune video generation. However, these approaches require decoding the video latent into RGB video as input for the reward model to compute the reward gradient. This process introduces significant memory costs, constraining efficiency. Moreover, while these methods primarily focus on enhancing overall quality or alignment with text prompts, none explicitly address the challenge of improving camera controllability in video generation. To address this gap, we propose a novel camera-aware 3D decoder specifically designed to enhance camera controllability in video diffusion models through reward feedback learning.

## 3 METHOD

We begin with a brief overview of camera-controlled video diffusion models, feed-forward Gaussian models, and reward feedback learning in Section 3.1. Section 3.2 describes the training of the camera-controlled video diffusion model with miscellaneous improvements. We then introduce our camera-aware 3D decoder in Section 3.3, followed by the meticulously designed reward feedback learning objective in Section 3.4. An overview of the entire framework is shown in Figure 2.

### 3.1 PRELIMINARIES

**Camera controlled video diffusion model** learns to model the conditional distribution $p(\mathbf{x}_0|c, \mathbf{s})$ of video tokens, where $\mathbf{x}_0$ denotes the video latent obtained from a video VAE (Yang et al., 2024c) , $c$ refers to the text or image condition and $\mathbf{s}$ is the camera condition. During training, noise $\epsilon_t$ is added to the latent $\mathbf{x}_0$ at each timestep $t \in [0, T]$ and a transformer model (Peebles & Xie, 2022) is optimized to predict this noise using the following objective:

$$L(\theta) = \mathbb{E}_{\mathbf{x}_0, \epsilon, c, \mathbf{s}, t} \left[ \|\epsilon - \hat{\epsilon}_\theta(\mathbf{x}_t, c, \mathbf{s}, t)\|_2^2 \right]. \tag{1}$$

Following prior methods (He et al., 2024; Bahmani et al., 2024a;b), we adopt the Plücker embedding (Sitzmann et al., 2021) as the camera condition, which provides pixel-aligned camera information and facilitates the use of ControlNet (Zhang et al., 2023) for conditioning.

**Feed-forward Gaussian model** aims to reconstruct 3DGS from a single image or multi-view images (Tang et al., 2024). It leverages a transformer-based architecture to project 2D images, along with their camera poses, into a pixel-aligned 3DGS. This 3D representation can then be differentiably rendered from arbitrary viewpoints, enabling photometric supervision and end-to-end optimization.

**Reward feedback learning** is a preference fine-tuning framework that directly optimizes the generation process using differentiable reward models and aims to improve the model by aligning the behavior of network output with external preference signals, such as human feedback or heuristic reward models (Wallace et al., 2024; Black et al., 2023; Xu et al., 2023).

### 3.2 ADDING CAMERA CONTROL TO VIDEO GENERATION

Following previous works (He et al., 2024; Bahmani et al., 2024b;a; Liang et al., 2024), we incorporate camera information (i.e., Plücker embeddings) into the denoising process through ControlNet (Zhang et al., 2023). The raw Plücker embeddings are first compressed along the spatial and temporal dimensions to align with the shape of the video latent, following the architectural design of Wonderland (Liang et al., 2024). To construct the ControlNet, we replicate the first several transformer blocks from the base video model and append a zero-initialized linear layer for stable training. Inspired by AC3D (Bahmani et al., 2024a), we copy the first several transformer blocks, which has been shown to strike a balance between controllability and computational efficiency. AC3D further observes that video diffusion models tend to establish low-frequency camera motion during the early stages of the denoising process. As a result, injecting camera control signals at later timesteps provides limited benefits and may even impair visual quality, rendering late-stage conditioning largely ineffective. Following this insight, we adopt a truncated normal distribution with a mean of 0.8 and a standard deviation of 0.075, restricted to the interval $[0.6, 1]$, to bias timestep sampling toward earlier denoising steps where camera control is most effective. The network architecture and training details can be found in the Appendix.

Despite these advancements, the overall camera controllability remains limited. Inspired by ReLF for video diffusion models, we aim to further improve the alignment by explicitly optimizing over denoising trajectories with reward feedback learning. To enable this, we first introduce a camera-aware 3D decoder that quantitatively evaluates the alignment of camera trajectory in the generated videos. This is followed by a dedicated reward objective for feedback learning.

### 3.3 CAMERA-AWARE 3D DECODER

ReFL methods primarily focus on enhancing visual quality or alignment with text through high-level semantic rewards, such as aesthetics evaluators or image-text similarity scores. However, existing models struggle to effectively assess how well the generated video matches the camera conditions.

A straightforward approach is to use COLMAP (Schönberger & Frahm, 2016), which can estimate camera poses from videos. Nonetheless, COLMAP demands substantial computational time to evaluate a single video and produces scale-invariant pose estimates, making it unsuitable for real-time training and pose supervision. Additionally, ReFL methods typically require decoding latent into RGB format for reward computation. This process is computationally expensive and only includes 2D information, whereas camera-controlled video generation inherently requires the video model to reason about 3D geometric information. Therefore, we aim to explore a 3D decoder for efficiently decoding video latent to video with 3D information incorporated. Moreover, the 3D decoder should also be camera-aware to ensure a quantitative assessment of the camera-video alignment.

To this end, we propose a latent-based feed-forward 3D Gaussian model as our camera-aware 3D decoder, which essentially extends video VAE to decode 3D representation. Specifically, we train a transformer that takes both video latent and their corresponding Plücker embeddings as input, and outputs per-pixel aligned 3DGS. The positions of these 3DGS are estimated by projecting the camera parameters together with the predicted ray distances $t$ using the relation $\mathbf{u} = \mathbf{r_o} + t \cdot \mathbf{r_d}$. To train the decoder, we randomly select a stride $s$ to sample a video sequence consisting of $T$ frames. The video VAE encoder first compresses these $T$ frames into latent, which are subsequently fed into trainable transformer blocks along with Plücker embeddings to predict the 3DGS. The training objective employs a combination of mean squared error (MSE) loss and LPIPS loss (Zhang et al., 2018a) between rendered images and ground-truth ones, ensuring both pixel-level accuracy and perceptual quality. To improve the rendering quality of unseen views and enforce 3D consistency, we additionally render novel views corresponding to the frames skipped during stride-based sampling. The detailed architecture can be found in the Appendix.

Within this framework, the camera poses play a crucial role in computing the final 3DGS, which act as input and projection variable. As a result, if the input latent and the camera poses are not well aligned, the 3D geometry deteriorates, leading to noticeably blurrier rendering. Based on this property, we design our reward for feedback learning.

## 3.4 Camera Reward Optimization

With the camera-aware 3D decoder, we propose Camera Reward Optimization (CRO) to use reward gradients to further improve the camera controllability. Considering the property that if the generated videos misalign with the camera condition, the renderings become blurry, a naive approach is to penalize the blurriness. However, directly penalizing the blurriness easily leads to reward hacking issues (Skalse et al., 2022). This means that the generated content may become clearer, but it may not align with the trajectory of the ground truth video. Hence, we regard minimizing the pixel-level difference between the rendered videos and ground-truth sequences as the reward. This design is consistent with the nature of the proposed camera-aware 3D decoder, which acts as a 3D representation decoder and supports rendering novel views, emphasizing low-level visual cues.

However, video generation introduces inherent stochasticity, making it infeasible to directly minimize the pixel-level difference with ground-truth videos since newly generated parts cannot align with ground-truth videos. To accommodate the stochastic nature, we adopt a visibility-aware reward strategy that restricts supervision to pixels that are visible in the conditioning image, which is generally deterministic. Visible mask can be derived by geometric warping, which requires depth information and camera poses. Fortunately, due to the inherent 3D structure of our camera-aware 3D decoder, we can obtain the rendered depth, which facilitates visibility estimation. Specifically, given the ground-truth video with corresponding camera poses $\mathbf{E} = [R; t] \in \mathbb{R}^{T \times 3 \times 4}$ and intrinsic matrix $K \in \mathbb{R}^{T \times 3 \times 3}$, and the image condition $\mathbf{I}_0 \in \mathbb{R}^{H \times W \times 3}$, which is the first frame of the video, we compute a per-frame visibility mask based on geometric warping. Omitting the temporal script, each pixel $(u, v)$ from the target view is back-projected into 3D world coordinates using the rendered depth map $\mathbf{D}$, intrinsic matrix $\mathbf{K}$, and camera extrinsic matrix.$\mathbf{E}$:

$$\mathbf{X}^{\text{world}}(u, v) = \mathbf{E} \cdot \left[ \mathbf{D}(u, v) \cdot \mathbf{K}^{-1}[u, v]^T \right]. \tag{2}$$

Next, the 3D points are projected into the conditioned reference view using the its extrinsic matrix $\mathbf{E}_0$ and intrinsic matrix $\mathbf{K}_0$. The projected 2D coordinates in the reference view are obtained by:

$$\mathbf{x}^{(0)}(u, v) = \mathbf{K}_0 \cdot \mathbf{E}_0^{-1} \cdot \left[ \mathbf{X}^{\text{world}}(u, v) \right]. \tag{3}$$

We then sample the reference depth map $\mathbf{D}_0$ at the projected location to obtain $D_0^{\text{proj}}(u, v)$. A visibility mask $\mathbf{M}$ is constructed by comparing the reprojected depth $\hat{z}^{(0)}(u, v)$ with the sampled

depth, and a pixel is considered visible if the two depths agree within a tolerance $\tau$:

$$M(u,v) = \begin{cases} 1, & \text{if } \left| \hat{z}^{(0)}(u,v) - D_0^{\text{proj}}(u,v) \right| < \tau \text{ and } D_0^{\text{proj}}(u,v) > 0, \\ 0, & \text{otherwise.} \end{cases} \quad (4)$$

With the visibility mask, we follow the VADER framework (Prabhudesai et al., 2024) and restrict the reward on deterministic pixels, defining a masked MSE loss and LPIPS loss between the rendered image $\hat{\mathbf{I}}$ and the ground-truth image $\mathbf{I}$ as:

$$\mathcal{L}_{\text{CRO}} = \mathcal{L}_{\text{MSE}}(\hat{\mathbf{I}}, \mathbf{I}, \mathbf{M}) + \lambda \cdot \mathcal{L}_{\text{LPIPS}}(\hat{\mathbf{I}}, \mathbf{I}, \mathbf{M}).$$

Here the ground-truth images include novel views that were skipped during stride-based sampling. The parameter $\lambda$ is set empirically to 0.5. Different from VADER, gradients propagate through all denoising time steps and our lightweight camera-aware 3D decoder can decode all $T$ frames.

## 4 EXPERIMENTS

### 4.1 DATASETS AND EVALUATION PROTOCOL

**Training Datasets.** Following previous methods (Bahmani et al., 2024b; Wang et al., 2024b; He et al., 2024), we utilized RealEstate10K (RE10K) (Zhou et al., 2018) as our training data, which contains approximately 65K videos in the train split. We used these 65K videos for training both the camera-aware 3D decoder and the camera-controlled video diffusion model.

**Testing Datasets.** Following previous works (Liang et al., 2024; Yu et al., 2024), we randomly selected 300 videos from the approximately 7K test sets of RE10K, ensuring no overlap with the training data. We also adopted the WorldScore (Duan et al., 2025) static benchmark for out of domain comparison, which consists of 2,000 static test examples.

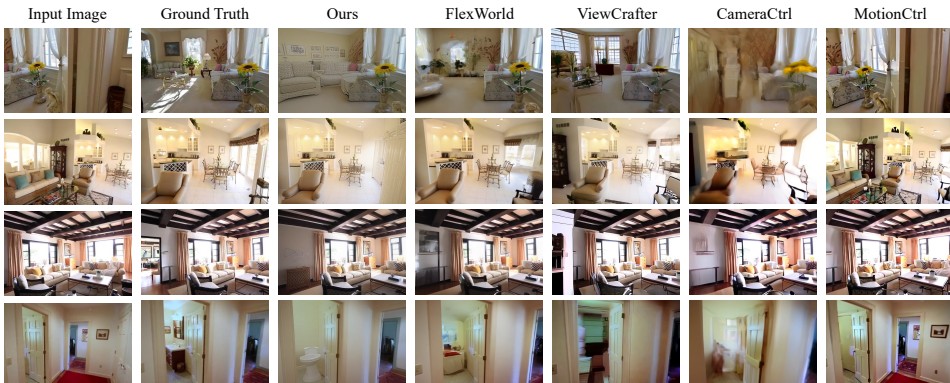

Figure 3: Qualitative comparison of video generation: our model produces novel views that are better aligned with the camera poses with higher quality, outperforming other methods.

**Evaluation Protocol.** We evaluated the quality of the generated videos using multiple metrics. Following previous works (Liang et al., 2024; He et al., 2024; Bahmani et al., 2024b;a), we employed Fréchet Inception Distance (FID) (Heusel et al., 2017) and Fréchet Video Distance (FVD) (Unterthiner et al., 2019) to assess visual quality. Additionally, PSNR, LPIPS, and SSIM metrics were used to evaluate the quality of novel view synthesis, camera controllability, and the performance of scene reconstruction. Following the approach in Wonderland (Liang et al., 2024), we also compute these metrics for the first 14 frames due to the randomness in generation. For further evaluating camera controllability, we used rotation error ($R_{\text{err}}$) and translation error ($T_{\text{err}}$) computed via DROID-SLAM (Teed & Deng, 2021) following WorldScore (Duan et al., 2025). Furthermore, we evaluated WorldScore (Duan et al., 2025) on WorldScore static benchmark. In addition, we compared the decoded video (from the video VAE decoder) and the rendered video (from the camera-aware 3D decoder) using the same generated latent. We report PSNR, SSIM (Wang et al., 2004), and LPIPS (Zhang et al., 2018b) as metrics to further evaluate the camera controllability.

## 4.2 IMPLEMENTATION DETAILS.

We built our model upon CogVideoX-5B-I2V (Yang et al., 2024c). To inject camera conditioning, we adopt ControlNet (Zhang et al., 2023), initializing the control branch with the first 8 base transformer blocks from the pretrained video model. For the camera-aware 3D decoder, we used 4 transformer blocks with a hidden dimension of 1,024. Please refer to Appendix for more details.

## 4.3 COMPARISON ON VIDEO GENERATION

We compared the proposed framework with four baselines: MotionCtrl (Wang et al., 2024b), CameraCtrl (He et al., 2024), ViewCrafter (Yu et al., 2024), and FlexWorld (Chen et al., 2025). The qualitative comparison is illustrated in Fig. 3, while the quantitative results are presented in Table 1. Our method surpasses existing approaches in both novel view synthesis and camera controllability.

| Method | Video Generation | | | | | | | 3D Scene Generation | | |
|---|---|---|---|---|---|---|---|---|---|---|
| | FID ↓ | FVD ↓ | $R_{err}$ ↓ | $T_{err}$ ↓ | PSNR ↑ | LPIPS ↓ | SSIM ↑ | PSNR ↑ | LPIPS ↓ | SSIM ↑ |
| Rec-only | - | - | - | - | - | - | - | 27.57 | 0.181 | 0.883 |
| MotionCtrl | 24.67 | 205.27 | 0.153 | 0.385 | 14.24 | 0.520 | 0.532 | 14.02 | 0.536 | 0.533 |
| CameraCtrl | 22.17 | 96.52 | 0.078 | 0.222 | 17.58 | 0.586 | 0.360 | 17.30 | 0.391 | 0.573 |
| ViewCrafter | 17.92 | 109.30 | 0.039 | 0.194 | 19.33 | 0.326 | 0.710 | 18.57 | 0.383 | 0.688 |
| FlexWorld | 17.23 | 103.94 | 0.030 | 0.177 | 21.27 | 0.292 | 0.731 | 19.12 | 0.360 | 0.703 |
| Ours | **11.22** | **81.35** | **0.023** | **0.152** | **23.77** | **0.226** | **0.766** | 21.72 | 0.272 | 0.717 |

Table 1: Quantitative comparison on video and 3D scene generation with the baseline methods.

## 4.4 COMPARISON ON SCENE GENERATION

To evaluate the effectiveness of our method for 3D scene generation, we compared the visual quality of the rendering results with the same four baseline methods using PSNR, LPIPS and SSIM between the renderings and ground-truth videos. To evaluate the upper bound of our camera-aware 3D decoder, we also reported the PSNR, LPIPS and SSIM between ground-truth video and rendered video (denoted as "Rec-only") using video and ground-truth camera pose as input. The quantitative results are reported in Table 1 and the qualitative comparison is illustrated in Fig. 4.

## 4.5 COMPARISON ON WORLDSCORE BENCHMARK

We also compared on the WorldScore static benchmarks (Duan et al., 2025). The quantitative results are reported in Table 2. Additional qualitative comparisons are in the Appendix. We reproduced the officially released code on this benchmark using the same test settings and hyperparameters.

## 4.6 ABLATION STUDY

We conducted an ablation study to validate the effectiveness of each component in our framework. The quantitative results are presented in Table 3, using PSNR, SSIM, and LPIPS metrics. These metrics compare the decoded video (from the video VAE decoder) and the rendered video (produced by the reward model) from the same generated latent, denoted as "Rendered vs Generated." Additionally, they compare the generated videos and rendered videos with the ground-truth ones, denoted as "Generated vs GT" and "Rendered vs GT," respectively.

**The effectiveness of reward feedback learning.** Reward feedback learning (ReFL) is crucial for enhancing the camera controllability. We compared the results before and after applying ReFL (denoted as "w/o ReFL") in Table 3. After implementing ReFL, the performance significantly improves, indicating that the reward gradients is effective and can further enhance camera controllability. We visualized a qualitative comparison and discuss further insights in the Appendix.

| Methods | WorldScore Average | Camera Control | Object Control | Content Alignment | 3D Consistency | Photometric Consistency | Style Consistency | Subjective Quality |
|---|---|---|---|---|---|---|---|---|
| MotionCtrl | 64.15 | 58.65 | 44.54 | 48.42 | 89.87 | 88.13 | 67.37 | 52.07 |
| CameraCtrl | 65.42 | 65.72 | 45.31 | 49.10 | 90.07 | 92.42 | 64.70 | 50.64 |
| ViewCrafter | 65.47 | 72.40 | 50.71 | 52.34 | 60.56 | 88.30 | 78.29 | 55.68 |
| FlexWorld | 71.35 | 68.16 | 56.15 | 53.66 | 84.43 | 91.31 | 86.07 | 59.65 |
| Ours | **74.45** | **86.26** | 49.75 | 46.46 | **90.64** | **93.30** | 89.78 | **64.95** |

Table 2: Quantitative comparison across control and consistency metrics. Higher is better.

Figure 4: Qualitative comparison of 3D scene generation: our model produces more photorealistic novel view rendering that are aligned with the camera poses, outperforming other methods

| Setting | Generated vs. GT | | | Rendered vs. GT | | | Rendered vs. Generated | | |
|---|---|---|---|---|---|---|---|---|---|
| Metric | PSNR↑ | LPIPS↓ | SSIM↑ | PSNR↑ | LPIPS↓ | SSIM↑ | PSNR↑ | LPIPS↓ | SSIM↑ |
| w/o ReFL | 21.57 | 0.282 | 0.720 | 18.93 | 0.361 | 0.642 | 24.34 | 0.231 | 0.798 |
| w/o visibility mask | 22.75 | 0.241 | 0.749 | 20.52 | 0.293 | 0.694 | 26.14 | 0.219 | 0.815 |
| w/o novel view | 22.88 | 0.232 | 0.756 | 20.88 | 0.279 | 0.706 | 26.45 | 0.202 | 0.824 |
| w/ CFG | 23.30 | 0.235 | 0.751 | 21.04 | 0.282 | 0.709 | 27.08 | 0.193 | 0.841 |
| Full model | **23.77** | **0.226** | **0.766** | **21.72** | **0.272** | **0.717** | **27.13** | **0.192** | **0.844** |

Table 3: Ablation study to validate the effectiveness of each component.

**The effectiveness of visibility mask.** The visibility mask plays a crucial role in accommodating the stochastic nature of generative models by supervising only the deterministic pixels in the conditioned image. We conducted an experiment without using the visibility mask (denoted as "w/o visibility mask") as shown in Table 3. The performance deteriorates without the visibility mask.

**The effectiveness of novel views.** Our camera-aware 3D decoder functions as a 3D decoder, projecting video latents into 3DGS. Unlike the video decoder, it can decode novel views in addition to the seen views that are input to the video encoder. This capability allows us to incorporate novel views as supervision. We conducted an ablation study to validate the effectiveness of using novel views, denoted as "w/o novel view" in Table 3. The performance of "w/o novel view" degrades, indicating the effectiveness of incorporating 3D geometric information.

**The effect of class free guidance.** During each denoising step, we have the option to use class-free guidance (CFG) or not. We conducted an ablation study to assess the impact of CFG on sampling. The qualitative comparison is presented in Table 3, labeled as "w/ CFG". The performance is comparable to that without CFG. However, since CFG results in twice the computational overhead during training, we have opted to disable CFG in our experiments.

## 5 CONCLUSION AND LIMITATION

**Limitation.** Despite the effectiveness of our proposed method, several limitations remain. First, the performance of the 3D decoder determines the upper bound of ReFL. For efficiency, we used only a 4 transformer blocks and trained solely on RE10K. Scaling up the network and dataset may further improve this upper bound. Second, 3DGS can only represent static scenes and is not suitable for dynamic scene reconstruction. Exploring 4DGS as a reward model is a direction for future work.

**Conclusion.** In this work, we investigate the problem of camera-controlled video diffusion models and 3D scene generation, where the quality heavily relies on the alignment between camera conditions and the generated videos. To further improve this alignment, we introduce a camera-aware 3D decoder for efficient decoding video latent to rendered videos for reward computation. During camera reward optimization, we propose to aligns the deterministic pixels between rendered videos and ground-truth videos. Extensive experiments validate the effectiveness of the proposed method, outperforming existing methods by a large margin.

## A  REPRODUCIBILITY STATEMENT

In our work, we have provided detailed descriptions of the training data, training parameters, and methodologies used in our experiments. We are committed to transparency and reproducibility in research. To this end, we will be releasing the corresponding code and datasets to the public in the near future. This will enable other researchers and practitioners to replicate our results and build upon our work, fostering an open and collaborative scientific community.

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

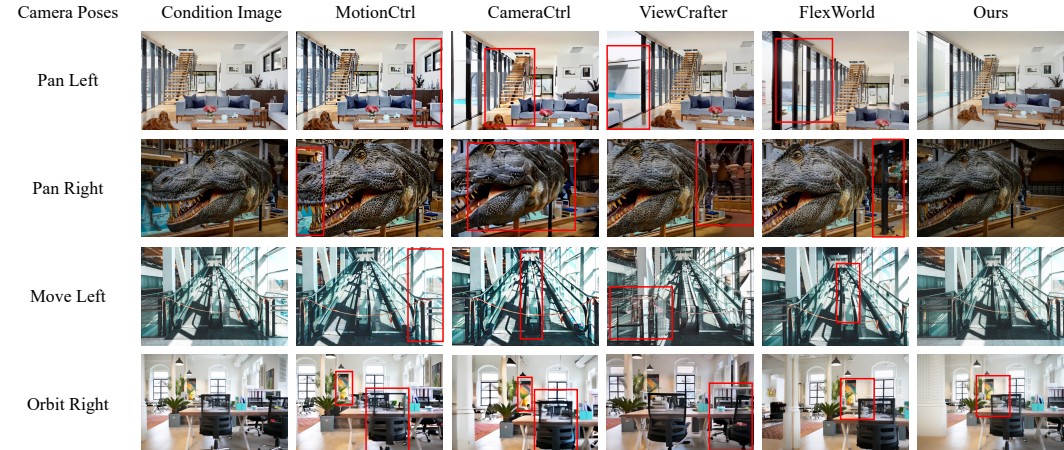

Figure 5: Qualitative comparison on WorldScore static benchmark.

# A  APPENDIX

## A.1  USE OF LLMS

I utilize Large Language Models (LLMs) to assist and enhance my writing process. While LLMs provide valuable support, we remain the primary authors and take full responsibility for the final output, ensuring it aligns with my personal style and meets ethical standards.

## A.2  QUALITATIVE COMPARISON ON WORLDSCORE STATIC BENCHMARK

We further visualize the qualitative comparison on the WorldScore static benchmark in Fig. 5. Our method generates more 3D consistent videos that match the given camera conditions.

## A.3  FURTHER ANALYSIS AND DISCUSSION

**Further Discussion on the Improvements of Using ReFL** We discuss more improvements of after ReFL. We visualized a qualitative comparison in Fig. 6. From the first case, we can observe that "w/ ReFL" maintains better photometric consistency during camera motion. In the results "w/o ReFL," there is an obvious photometric shift. Our camera-aware 3D decoder leverages 3DGS to represent the scene, which is typically photometrically consistent across novel views. This property is also distilled into the video diffusion model by ReFL, which is favorable for this task. Moreover, we found that "w/ ReFL" can effectively suppress dynamic generation, maintaining better 3D consistency in generated videos. Since 3DGS is essentially a static 3D representation, this property is also distilled into the video model to produce content that is both static and 3D consistent. The corresponding video can be found in the Supplementary Materials.

**The scale of camera conditions**. Although the camera poses in RE10K are normalized to a unified scale as described in (Zhou et al., 2018), we observed that there are still variations in scale within this unified framework. Specifically, some movements are more pronounced while others are subtler. During inference, we found that by manipulating the scale of the camera conditions, our model can effectively perceive these scale variations and generate videos that accurately reflect the intended degree of movement. We visualized some examples in Fig. 7, where the same image was used as a condition, but the scale of the camera pose was varied for each generation.

**The choice of Plücker embeddings as conditions**. Recent camera-conditioned video generation methods can be roughly divided into two categories: those that use point cloud renders as conditions and those that use Plücker embeddings as conditions. We chose Plücker embeddings due to their flexibility and generalization capabilities. However, our method is general and can also be employed in frameworks where point cloud renders are used as conditions. Using point cloud renders as conditions typically relies on external models (Wang et al., 2024a; 2025) for simultaneous point

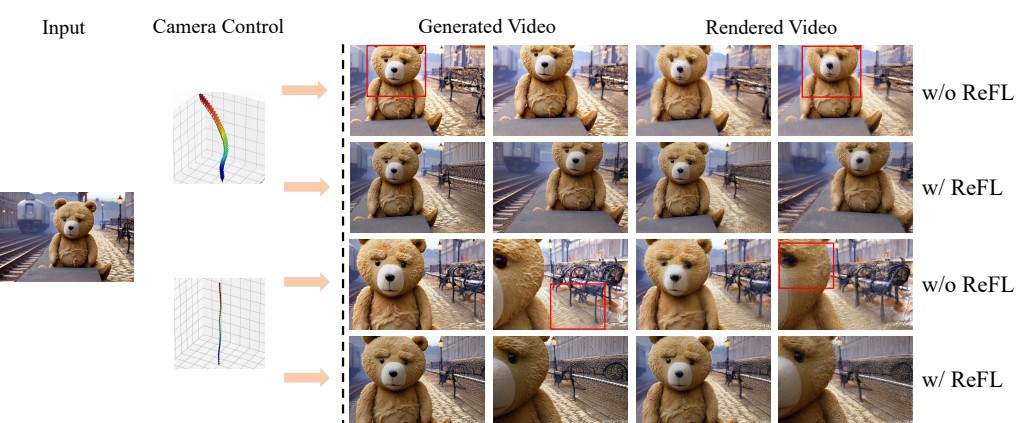

Figure 6: Qualitative comparison between "w/o ReFL" and "w/ ReFL".

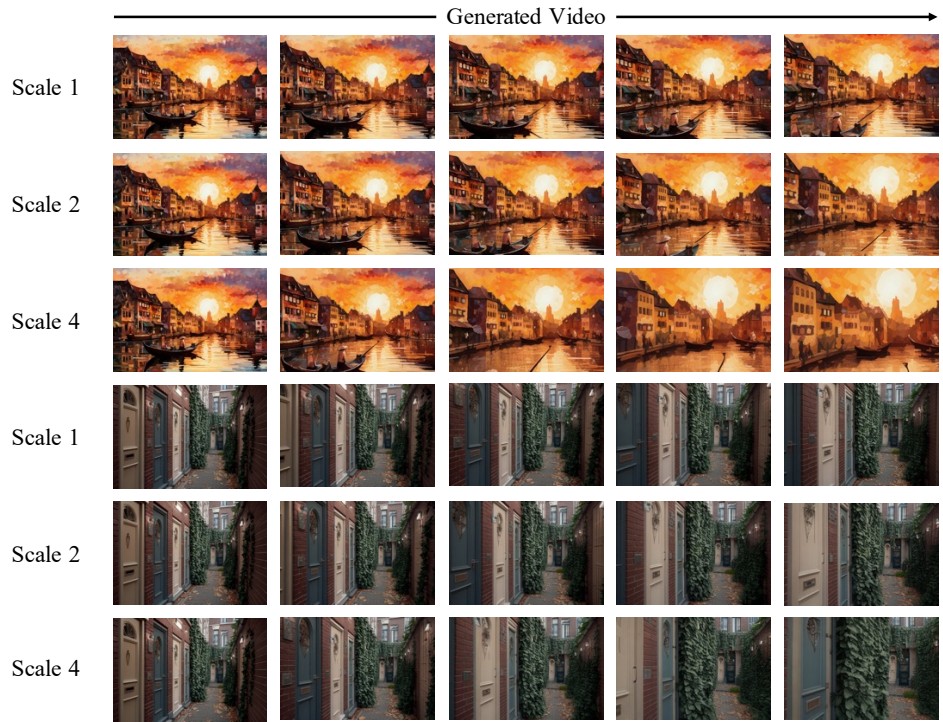

Figure 7: Our model is capable of perceiving scale variations and generating videos that accurately reflect the intended degree of movement. A larger scale results in more pronounced movements.

Input Image     Point cloud renders     Generated frame

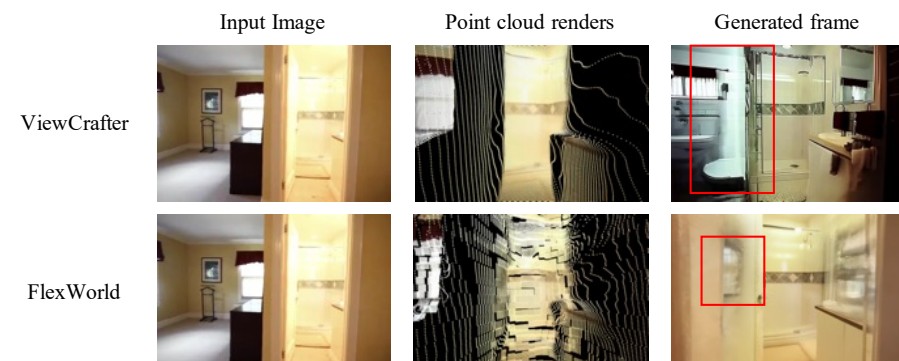

Figure 8: Using point cloud renders as camera condition incur a rendering leakage problem, affecting the quality of novel view synthesis.

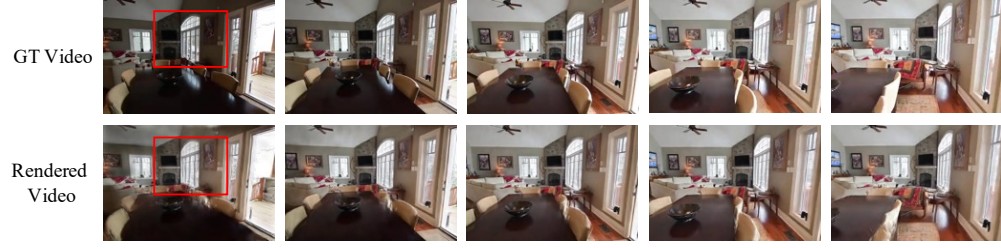

Figure 9: An example of ground-truth videos with varying exposure levels. The rendered video from 3DGS tends to exhibit an average exposure, which differs from the ground-truth video.

cloud and camera pose estimation to achieve alignment. If a dataset contains ground-truth metric camera poses, the estimated point cloud should be further processed to align with the ground-truth poses, while Plücker embeddings can be easily obtained without any preprocessing. Moreover, point cloud renders incur a rendering leakage problem: as the camera view changes, background points may be incorrectly rendered into the foreground due to improper handling of occlusion relationships, affecting the realism and consistency. We show an example in Fig. 8.

**The reconstruction performance of camera-aware 3D decoder.** Our camera-aware 3D decoder is exclusively trained on the RE10K dataset, which comprises estate videos exhibiting varying exposure changes as the camera perspective shifts. The model generates per-frame 3DGS and uses them as a global 3D representation for rendering. However, exposure changes result in variations in the predicted spherical harmonics, which can degrade rendering quality to some extent. We show some examples in Fig. 9.Collecting more consistent videos with precise camera poses can further enhance the reconstruction performance of the camera-aware 3D decoder.

A.4    THE EFFICIENCY OF CAMERA-AWARE 3D DECODER

We compared the efficiency of our proposed camera-aware 3D decoder and video VAE decoder in terms of GPU memory cost and time cost, as shown in Table 4. When using the video VAE decoder, we can only decode 2 temporal latents in each iteration with 80GB of GPU memory during ReFL training, while camera-aware 3D decoder can decoder all 49 frames. Moreover, the visibility mask is not available with video VAE decoder.

Table 4: Comparison of GPU Memory and Time Cost

| Decoder Type | GPU Memory Cost (GB) | Time Cost (s) |
| --- | --- | --- |
| Camera-aware 3D Decoder | 8.44 | 0.559 |
| Video VAE Decoder | 43.17 | 5.602 |

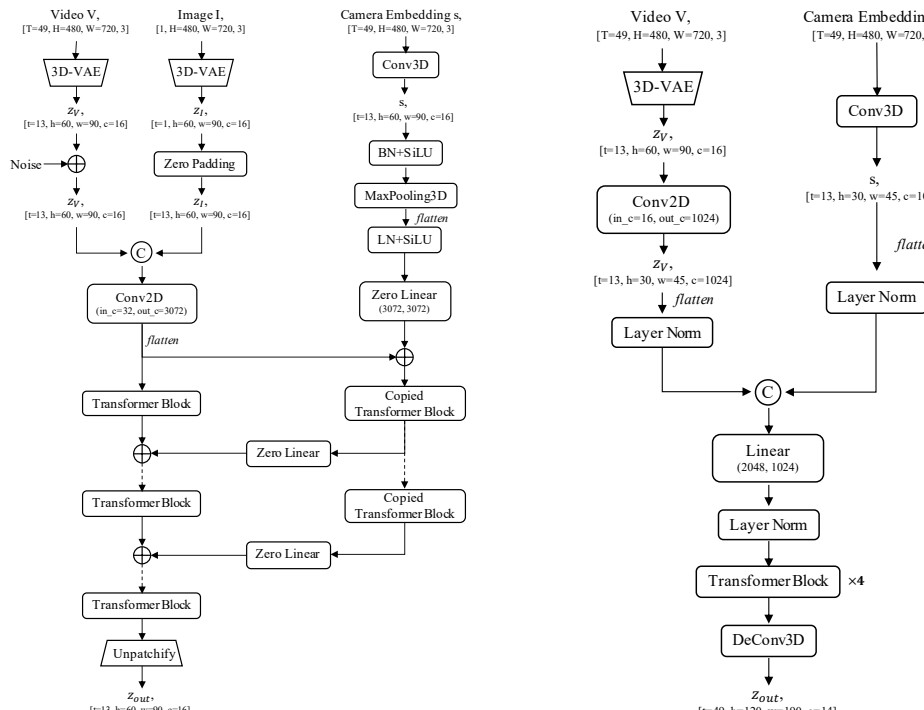

Figure 10: The detailed network architecture for camera-controlled video diffusion model.

Figure 11: The detailed network architecture for camera-aware 3D decoder.

## A.5 OPTIMIZATION AND ADDITIONAL MODEL DETAILS

**Optimization Details.** We used the Adam optimizer (Kingma & Ba, 2014). In the first stage, the learning rate was set to $1 \times 10^{-4}$. In the second stage, the learning rate was set to $3 \times 10^{-4}$, and in the third fine-tuning stage, the learning rate was set to $1 \times 10^{-5}$. In the first stage, we used 16 NVIDIA A800 GPUs for basic camera-controlled video model training with a batch size of 16 for 10K steps. In the second stage, we used 32 NVIDIA A800 GPUs to train our camera-aware 3D decoder with a batch size of 32 for 100K steps. In the third stage, we used 16 NVIDIA A800 GPUs for reward-based feedback learning with a batch size of 16 for 5K steps. In this stage, we perform denoising a total of 7 times, and the reward gradient propagates through all the denoising steps.

**Network architecture.** Our network architecture is similar to that of Wonderland (Liang et al., 2024). The details of the network for the first stage are shown in Fig. 10. Pixel-aligned Plücker embeddings are compressed via a Conv3D layer, ensuring the camera latent shares the same dimension with the video latent. Then, batch normalization, an activation layer, and a max pooling layer are used to convert the camera latent into sequential tokens as ControlNet input. For efficiency considerations, we only copied the first 8 transformer blocks.

For the camera-aware 3D decoder, we elaborate on the network architecture in Fig. 11. We convert the video latent using Conv2D into visual tokens. To ensure the same dimension for the camera embedding, we leverage Conv3D for spatial-temporal compression. Then, visual tokens and camera tokens are concatenated along the channel dimension. Four Transformer blocks and a DeConv3D layer are used to process the concatenated tokens into pixel-aligned 3DGS. Note that we do not recover the original spatial resolution for 3DGS, which we found is sufficient to represent a scene. During training, we employed 49 supervision views, where 14 frames are randomly sampled from the source video clip as seen views, and the remaining 35 are selected from disjoint frames as unseen views to ensure 3D consistency

### A.6    PLÜCKER EMBEDDINGS DERIVATION

Given a camera trajectory with extrinsic parameters $\mathbf{E} = [\mathbf{R}; \mathbf{t}] \in \mathbb{R}^{3\times4}$ and intrinsic matrix $\mathbf{K} \in \mathbb{R}^{3\times3}$, we derive the Plücker representation $\mathbf{s} = (\mathbf{o} \times \mathbf{d}', \mathbf{d}')$ for each pixel $(u, v)$. The camera's world-space origin $\mathbf{o}$ is defined by the translation vector $\mathbf{t}$. The direction vector $\mathbf{d}$ from the camera center to the pixel is computed as:

$$\mathbf{d} = \mathbf{R}\mathbf{K}^{-1}[u, v, 1]^T + \mathbf{t}$$

where $\mathbf{K}^{-1}[u, v, 1]^T$ transforms the pixel coordinates into normalized camera coordinates, and $\mathbf{R}$ rotates these coordinates into the world space. The unit-normalized direction $\mathbf{d}'$ is obtained by normalizing $\mathbf{d}$:

$$\mathbf{d}' = \frac{\mathbf{d}}{\|\mathbf{d}\|}$$

The Plücker representation $\mathbf{p}$ is then given by:

$$\mathbf{s} = (\mathbf{o} \times \mathbf{d}', \mathbf{d}')$$

where $\mathbf{o} \times \mathbf{d}'$ represents the moment of the line, calculated as the cross product of the camera origin and the unit direction vector. We generate a per-frame Plücker tensor $\mathbf{P}_i \in \mathbb{R}^{6\times h\times w}$, ensuring that its spatial dimensions $h$ and $w$ align with those of the video, which is favorable for conditioning with ControlNet.

### A.7    PROJECTION FORMULATION FOR THE MEAN OF 3DGS

In this section, we describe how the XYZ positions of the 3DGS are obtained through Plücker embedding. Plücker embedding defines the ray origin and direction for each pixel, allowing us to map the network's output depth to spatial coordinates.

The Plücker embedding provides a representation of lines in 3D space using two vectors: the ray origin $\mathbf{o}$ and the ray direction $\mathbf{d}$. For each pixel, these vectors define a line in space. The depth value $z$ output by our network can be used to compute the XYZ position $\mathbf{p}$ of the 3DGS using the following mapping formula:

$$\mathbf{p} = \mathbf{o} + z \cdot \mathbf{d}$$

Here, $\mathbf{o}$ is the origin of the ray, $\mathbf{d}$ is the direction of the ray, and $z$ is the depth value. This formulation allows us to convert depth information into precise spatial coordinates, effectively reconstructing the 3D geometry of the scene.

By leveraging Plücker embedding, our approach ensures that each pixel's depth is accurately projected into 3D space, facilitating the generation of a pixel-aligned 3DGS representation. However, if the generated video latent does not match the camera condition, the projection may lead to degraded geometry, which further affects the rendering quality.

### A.8    THE EFFECT OF MISMATCHED CAMERA POSE

During our camera-aware 3D decoder training, we pair the video latent with the ground-truth camera pose as input. It is crucial to describe the scenario where the input pose does not align with the camera motion in the video. Since the pose serves as both the network input and a key projection parameter during the camera-aware 3D decoder training, any inconsistency can lead to blurred rendering effects. As illustrated in the Fig. 12, when perturbations are added to the ground-truth pose, the rendered images become noticeably blurred.

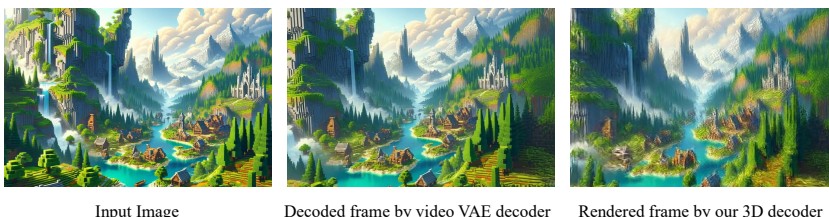

Input Image      Decoded frame by video VAE decoder      Rendered frame by our 3D decoder

Figure 12: We add perturbation to the given camera pose, and the rendered image becomes noticeably blurred, indicating the importance of aligned poses for rendering photorealistic images.

