# OpenReview forum: "CamPilot: Improving Camera Control in Video Diffusion Model with Efficient Camera Reward Feedback"
_ICLR.cc/2026/Conference — ICLR 2026 Conference Withdrawn Submission_

### Official Review · Reviewer_qxx8 · 2025-10-15

**Soundness:** 2
**Presentation:** 2
**Contribution:** 2
**Rating:** 4
**Confidence:** 5

**Summary:**

This paper proposes CamPilot, a technique to improve camera control in video diffusion models. Concretely, this work uses Reward Feedback Learning (ReFL) to improve the camera control. For this, a 3DGS decoder on top of the video latents is used to directly generate 3D Gaussians instead of RGB videos. Then, the renderings of the 3D Gaussians are used for reward feedback. Using the depth to obtain a visibility mask, the generated output is masked out to only supervise on input content.

**Strengths:**

- Novelty of using reward feedback for camera control: I have not seen a work that uses 3DGS to improve the underlying video model and its 3D consistency/camera control precision. The direction seems promising.

**Weaknesses:**

- Camera-aware 3DGS decoder already proposed: The first point of the contribution list claims the camera-aware 3DGS decoder to be a contribution. However, the approach just follows Wonderland [1] without changes. The claim of the whole pipeline seems to be a big overclaim and the authors should have acknowledged Wonderland a lot more during the paper. Wonderland is mentioned and referenced in the paper, but there are no differences in the approaches. The main point of the paper is the reward-based feedback learning, not the 3DGS decoder.
- Supplementary video presentation: The supplementary video presentation is one of the key components of a video generation submission. But there is no website but only separate videos and it is not clear what is happening. Moreover, there are no side-by-side comparisons with previous works. You should also show multiple generated scenes for one camera trajectory to show that they are consistent and the camera control works across scenes.
- Unclear handling of dynamic objects: Currently, there is no guarantee that the scene remains static. While most results are designed for scenes without objects in them, that could move, the approach seems to not handle dynamic objects which would get animated by the video model in normal cases. Those animated objects would then lead to blurry 3DGS outputs and bad reward feedback. Hence, currently the model is restricted to static scenes.

[1] Liang et al., Wonderland: Navigating 3D Scenes from a Single Image, CVPR 2025

**Questions:**

I generally like the idea of using 3DGS output to improve 3D consistency/camera control precision of video models. However, the majority of the paper overclaims the architectural contributions since they just follow Wonderland. The main formulation of the reward feedback is simple, which is good, but it is not analyzed that deeply. Moreover, the supplementary material is not organized well and is difficult to digest. I highly recommend preparing a supplementary website with side-by-side comparisons.

I would like authors to address the following question:

- What are the architectural contributions described in Sec. 3.1-3.3? All the parts mentioned are just reusing other works.

I am currently negative but happy to see what the authors say about the actual contributions of the work.

---

### Official Review · Reviewer_n4a2 · 2025-10-17

**Soundness:** 2
**Presentation:** 1
**Contribution:** 3
**Rating:** 4
**Confidence:** 3

**Summary:**

This paper introduces CamPilot, a video diffusion framework designed to improve camera controllability through a reward feedback learning strategy. The authors propose a camera-aware 3D decoder that decodes video latents into 3D Gaussian representations to evaluate camera-video alignment efficiently. The model achieves better camera control and 3D consistency compared to existing methods on the RealEstate10K and WorldScore benchmarks.

**Strengths:**

1. Using 3DGS to enhance camera-guided video generation is a good starting point

**Weaknesses:**

1. The writing quality of this paper needs improvement. Many expressions are verbose and not concise enough. For example, Sections 2.2 and 2.3 contain multiple repetitions of earlier content, resulting in unnecessary length. Moreover, the core comparison experiment showing how much computational cost is reduced compared to VAE decoding is placed only in Appendix A.4. From the main text alone, the experimental details are unclear.

2. The baseline methods used for comparison are somewhat outdated. The paper lacks comparisons with several recent and highly relevant approaches, such as CamI2V [1], OmniCam [2], RealCam-I2V [3], and ReCamMaster [4].

3. The proposed method employs a multi-stage reward scoring strategy for evaluating camera trajectories. Although the authors claim that this method is efficient, I am curious whether it could lead to error accumulation across stages. Compared with approaches that estimate camera trajectories from decoded videos (e.g., via VAE decoding) and directly compare them to ground-truth trajectories, how large is the accuracy gap between these two strategies?

4. Minor revision suggestions:  Please correctly use the `\citep` command for references, e.g., Line 208 for *UniFL*.

[1] CamI2V: Camera-Controlled Image-to-Video Diffusion Model

[2] OmniCam: Unified Multimodal Video Generation via Camera Control

[3] RealCam-I2V: Real-World Image-to-Video Generation with Interactive Complex Camera Control

[4] ReCamMaster: Camera-Controlled Generative Rendering from A Single Video

**Questions:**

See WeakNess

---

### Official Review · Reviewer_xyF9 · 2025-10-31

**Soundness:** 3
**Presentation:** 3
**Contribution:** 3
**Rating:** 4
**Confidence:** 3

**Summary:**

This paper improves camera control in video generation using a camera-aware 3D decoder and Reward Feedback Learning (ReFL). The decoder maps video latents to 3D representations to assess alignment. ReFL then optimizes this alignment using a novel visibility-aware reward that supervises only visible pixels. Experiments on RealEstate10K and WorldScore show significant improvements in camera control and visual quality.

**Strengths:**

1. Originality
This paper addresses the under-explored problem of enforcing camera conditioning in video diffusion models using ReFL, thereby improving alignment between generated footage and prescribed camera parameters.

2. Quality
The proposed approach incorporates a camera-aware 3D decoder that efficiently evaluates video–camera consistency while reducing computational overhead. Experimental results demonstrate clear gains in both camera control accuracy and overall visual quality.

3. Clarity
The presentation is well organized and easy to follow. Detailed diagrams—particularly the architectural overview in Figure 2—effectively convey the components and workflow of the framework.

4. Significance
By enhancing camera control in video generation, this work has immediate relevance to applications such as virtual reality and robotics. Moreover, it delivers an efficient feed-forward 3D reconstruction module and addresses the efficiency bottlenecks of reward-based learning in diffusion models.

**Weaknesses:**

1. The paper provides limited insight into how the method scales to larger datasets or real-world deployments; a more detailed analysis of computational requirements and potential bottlenecks would strengthen its practical relevance.
2. By relying on 3DGS, the approach is inherently restricted to static scenes, as the authors acknowledge, limiting its applicability to dynamic or non-rigid environments.
3. The pixel-level reward signal may be too low-level to capture high-order semantics, potentially leading to overly smoothed outputs that sacrifice semantic fidelity.

**Questions:**

1. How general is the proposed framework? How well does it perform when applying the camera-aware 3D decoder and CRO to other baseline models, such as MotionCtrl or ViewCrafter?
2. The claim that latent-pose misalignment causes blurry renders is central to the reward design , supported qualitatively by Fig. 12. Could the authors provide quantitative validation, such as a plot showing rendering quality (PSNR/LPIPS) degrading as camera pose noise increases?
3. Since the visibility mask only supervises regions from the first frame, how does the model ensure 3D consistency in large, newly revealed areas that receive no reward gradient? Have any artifacts been observed in these un-supervised regions?
4. Can the authors quantify the sensitivity of the final model's performance to the 3D decoder's quality? For example, what is the expected performance gain for every 1-point PSNR improvement in the decoder?
5. What is the justification for using 7 denoising steps for reward computation, and how does this hyperparameter affect the trade-off between performance, stability, and cost?
6. How does the initial training's timestep sampling bias interact with the subsequent reward optimization stage? Is the reward gradient also biased towards certain timesteps?
7. Given the model's scene-centric training on RealEstate 10K and its static scene design, how does it perform on object-centric generation tasks?

---

### Official Review · Reviewer_62yC · 2025-10-31

**Soundness:** 3
**Presentation:** 2
**Contribution:** 2
**Rating:** 6
**Confidence:** 4

**Summary:**

This paper introduces CamPilot, a novel framework aimed at enhancing camera controllability in video diffusion models through reward feedback learning. The authors identify a persistent challenge in aligning generated video content with specified camera trajectories, which undermines 3D consistency in downstream tasks such as scene reconstruction. To address this, they propose a camera-aware 3D decoder that projects video latent and camera poses into 3D Gaussians (3DGS), enabling efficient rendering and reward computation. The framework is evaluated on RealEstate10K and WorldScore, demonstrating improved camera alignment and visual fidelity.

**Strengths:**

1. Introduces a feed-forward 3D Gaussian-based decoder that efficiently evaluates camera-video alignment without reliance on computationally intensive post-processing tools like COLMAP.
2. Applies reward feedback learning (ReFL) to optimize camera adherence, which represents a previously underexplored direction in video diffusion.
3. Enables high-quality 3D scene reconstruction directly from video latents and camera poses, bypassing computationally expensive per-scene optimization.

**Weaknesses:**

1. The evaluation is limited to static scene datasets, this potentially limits its applicability for real-world video generation tasks.
2. The method assumes precise extrinsic and intrinsic camera parameters are available, which may not be a valid assumption in real-world applications.

**Questions:**

1. How does the method handle dynamic scenes involving object motion or non-rigid transformations? It is recommended that the authors include evaluation results evaluated on open-sourced dynamic RealCam-VID[1].
2. It is recommended that more comparison with other recent methods focused on camera control be added, such as RealCam-I2V[2]. We recommend including quantitative and qualitative results on the open-sourced dynamic RealCam-VID[1] dataset, alongside an analysis detailing the advantages of the proposed CamPilot framework in final paper.
3. How well does the model perform under conditions of rapid camera motion or large trajectory shifts (e.g., simulated drone flights or rapid rotations)? The authors are encouraged to provide quantitative or qualitative results in such scenarios.
4. How is camera scale handled during training and inference? The authors are recommended to clarify whether the model rely on absolute or relative scale inputs, and how does it ensure scale consistency across scenes?

[1] RealCam-Vid: High-resolution Video Dataset with Dynamic Scenes and Metric-scale Camera Movements

[2] RealCam-I2V: Real-World Image-to-Video Generation with Interactive Complex Camera Control

---

### Note · Authors · 2025-11-12

I have read and agree with the venue's withdrawal policy on behalf of myself and my co-authors.